# Evaluation of the COVID-19 Era by Using Machine Learning and Interpretation of Confidential Dataset

Andreas Andreou [1,*], Constandinos X. Mavromoustakis [1], George Mastorakis [2], Jordi Mongay Batalla [3] and Evangelos Pallis [4]

1 Department of Computer Science, University of Nicosia and University of Nicosia Research Foundation (UNRF), Nicosia 1700, Cyprus; mavromoustakis.c@unic.ac.cy

2 Department of Management Science and Technology, Hellenic Mediterranean University, 72100 Agios Nikolaos, Greece; gmastorakis@hmu.gr

3 National Institute of Telecommunications, Warsaw University of Technology, 04894 Warsaw, Poland; jordi.mongay.batalla@pw.edu.pl.com

4 Department of Industrial Design and Production Engineering, School of Engineering, University of West Attica, 12243 Athens, Greece; epallis@uniwa.gr

* Correspondence: andreou.andreas@unic.ac.cy; Tel.: +357-96536499

**Abstract:** Various research approaches to COVID-19 are currently being developed by machine learning (ML) techniques and edge computing, either in the sense of identifying virus molecules or in anticipating the risk analysis of the spread of COVID-19. Consequently, these orientations are elaborating datasets that derive either from WHO, through the respective website and research portals, or from data generated in real-time from the healthcare system. The implementation of data analysis, modelling and prediction processing is performed through multiple algorithmic techniques. The lack of these techniques to generate predictions with accuracy motivates us to proceed with this research study, which elaborates an existing machine learning technique and achieves valuable forecasts by modification. More specifically, this study modifies the Levenberg–Marquardt algorithm, which is commonly beneficial for approaching solutions to nonlinear least squares problems, endorses the acquisition of data driven from IoT devices and analyses these data via cloud computing to generate foresight about the progress of the outbreak in real-time environments. Hence, we enhance the optimization of the trend line that interprets these data. Therefore, we introduce this framework in conjunction with a novel encryption process that we are proposing for the datasets and the implementation of mortality predictions.

**Keywords:** urban; antiviral; post-COVID-19; spatial distancing; machine learning; sustainable

## 1. Introduction

Promoting the early detection of an outbreak is paramount for the sustainable development of antiviral urban ecosystems. A prerequisite is to build an antiviral intelligent city framework in a multigenerational urban environment relative to the post-COVID-19 era. Humanitarian efforts in the pandemic's framework deployed novel technological solutions based on the Internet of Things (IoT), machine learning, cloud computing and artificial intelligence (AI). Through our research study, we aim to contribute knowledge-based solutions for the direct control of the exponential promotion of cumulative infectious cases and the cumulative amount of mortality due to COVID-19. We propose an innovative system that could accurately forecast the progress of a virus spread and inform governments to align their policies against the outbreak in real-time. The main objective is to elaborate on the aspects that could construct a sustainable and effective strategy against disease outbreaks and an intelligent urban ecosystem based on technological initiatives.

The novel severe acute respiratory syndrome coronavirus 2, temporarily named SARS-CoV-2 and permanently renamed by WHO on 11 February 2020 as Corona Virus

Disease 2019 'COVID-19', caused enormous adverse consequences worldwide. To control spread, countries aligned their policies according to spatial distancing among citizens. However, various technological innovations and response initiatives have been developed to handle the unprecedented situation. It is paramount to establish a reference framework to contribute to effective defences against viruses and reliable urban ecosystems. Therefore, a multigenerational framework will be a significant step in improving digitalization and responding to post-COVID 19 antiviral society [1]. Fundamental mathematics is paramount to understanding the pandemic's progress, as we can interpret and forecast cumulative infectious cases [2]. The development of machine learning techniques to predict the current situation or future outbreaks in conjunction with cloud computing will benefit the timely assessment of the epidemiological portrait. We first interpret the data from cumulative infectious cases and deaths due to COVID-19 and compare their trends. Then, we evaluate the three constructed models, linear, exponential and polynomial, by using R-squared to determine which model best fits. We propose improving the best fit predictive model by using the implementation of machine learning techniques. Real-time data will be driven for evaluation from the cloud repository, which will secure the datasets with the proposed fragmentation scheme [3,4]. The objective is to accurately predict the curve's progress for governments to implement their policy reform from an early stage of the outbreak. We also apply hypothesis testing regarding Italy's monthly mortality rate. We analyse the cumulative infected cases compared to the daily rate of patients derived from the Polymerase Chain Reaction (PCR) tests during the pandemic's second wave.

COVID-19, an extremely contagious disease, was first reported in the Wuhan, Hubei Province, China, and affected a vast percentage of the world's population. After thirteen months, WHO reported that 95,623,389 people have been infected, and 2,042,644 deaths occurred due to the pandemic. The inversely proportional fact of the vast number of infectious cases associated with the limited interval of time interprets the cumulative infectious cases as exponential due to the curve's rapid increment. Fatality rates statistically prove that mortality occurs mainly in older adults and patients who suffer from chronic diseases with a weakened immune system. Thus, the lack of vaccination has led governments to implement national lockdown rules in order to restrict the spread as much as possible and to respond to healthcare needs. Communities are also taking technological innovation initiatives to deter the pandemic's waves. Edge computing could contribute several novel ideas to thwart the spread. The development of machine learning, in conjunction with Cloud Computing, is of paramount importance. For instance, the projected alert for the future increase in new infectious cases in the community is a real weapon against the invisible enemy. We can tailor quarantine policies and restrictions accordingly.

The interpretation of cumulative infectious case data through fundamental mathematics includes linear, exponential and polynomial regression models. By generating R-squared by Microsoft Excel, we conclude that sixth-degree polynomial goodness of fit assesses numeric measures accurately as the discrepancy between observed values and the values expected under the model is limited. We calculated the coefficients through the least squares method in order to reduce the variance between the values generated from the sixth-degree polynomial that interprets data and the initial dataset. Therefore, we can forecast the outbreak's progress by expanding the curve. In addition to that, we can derive the inflexion points by using the second differentiation of the function. They can predict the cavity of the curve and, thus, forecast the increment or decrement of the cases.

Using data driven from a cloud repository called "Our World in Data" [5], our objective is to develop a novel forecasting model based on fundamental mathematics that can interpret daily reports in real-time. In conjunction with cloud computing, machine learning deployment enables data procurement from a real-time data repository to predict the virus's course. Accurate predictions of infected cases from the virus will allow governments to adjust their policies because the system will inform them about the maximum number of patients, the number of total infected citizens and the expected period during which the pandemic will last. The mathematical framework will be informed daily, and the

polynomial function will be recomputed. The goal is to have an up-to-date curve for accurate forecasts.

One of the most effective cryptographic methods that enable secure data exchange is fragmentation [6,7]. During the COVID-19 period, reliability in data acquisition infrastructure was a prerequisite. Therefore, we thoroughly elaborated the fragmentation method by utilizing potential datasets from COVID-19 cases.

Furthermore, the security and reliability of data-sharing infrastructure need a community of trust. Therefore, this paper also introduces an encryption frame based on data fragmentation.

We statistically analysed Italy's monthly mortality rate to forecast the fatality rate that corresponds to cumulative infectious cases in cumulative cases. Moreover, we studied the contradiction between the cumulative infectious cases' forecast and the PCR testing rate from the pandemic's second wave. More specifically, if we interpret cumulative infectious cases and the daily percentage of newly infected patients from the total daily number of examinations, we end up with a different prediction.

## 2. Literature Review

In order to determine how the Dark Web has been influenced by recent global events, such as the pandemic situation Razaque et al. studied with the usage of a crawler, which scans the network and collects data for further analysis with machine learning [8]. The pandemic, along with its conservatism measures, has become the new norm for human life. Yaxi et al., based on data acquisition regarding mobile phone positioning of thirty-one million users in Beijing, China, tracked vicissitudes in two rudimentary human daily activities: dwelling and working. They concluded that working concentrations decreased approximately 60% urban wide during the pandemic outbreak while dwelling decreased about 40% [9]. Andreou and Mavromoustakis et al. proposed a cloud-based framework for accurately identifying truly positive infectious cases. Moreover, they introduced a novel solution aiming to prevent and control the outbreak based on smartphones and initiatives within a Naive Bayesian Network (NBN). In addition, they sought to provide local health authorities with a risk assessment of geolocation risk and early findings to trigger them to increase test rates in high-risk areas [1]. An approach around economic management orientation to improve the accuracy of the forecast for the pandemic was proposed by Xuan et al. based on a self-correcting intelligent pandemic prediction model [10]. For the same purpose, Rongbo and Qianao et al. have introduced a real-time warning model that studies the factors of public opinion on the internet and the dynamic characteristics of epidemic incidents. Therefore, they constructed a vector machine and a logistic regression model in order to enhance the prediction based on COVID-19 data [11]. Following the same motive, Srikanta et al. have analyzed COVID-19 mortality and infectious diseases in Europe using spatial regression models. More specifically, they select thirty-one countries for modelling and consequent analysis [12]. Elbasi et al., aiming to discover vulnerable groups and to reduce the impact of the disease on particular groups, have deployed machine learning techniques.

Naglaa and Ehab et al. stated that "architecture and urbanism after the COVID19 epidemic will never be the same" and they might be correct [13]. Their scope was to research the current pandemic situation in order to enhance the response to future similar outbreaks. Chanjuan and Zhiqiang et al., aiming to prevent the spread of corona-virus, elaborated on research fields with respect to spatial distance and indoor ventilation efficiency [14]. Based on the same research area, Antony, Velraj and Fariborz et al. studied the spread of COVID-19 under several different climates and environmental conditions: indoor and outdoor [15]. To overcome several lockdown policies' adverse economic impact due to continually pandemic waves, this paper [16] proposed a real-time data-driven dynamic clustering framework. Finally, Xing-Yi et al. examined the risk of Coronavirus spreading to health and care ecosystems in order to enhance sustainable work in the hospital environment [17].

Cities across the EU need to develop supportive environments that provide access to a range of facilities and services to achieve a higher quality of life for their senior citizens in order to improve the future after the Coronavirus pandemic and to confront the fear of a potential upcoming pandemic resurgence. A prerequisite is to identify the areas where cutting-edge technology could be integrated and be beneficial to ageing societies. The three main fields we need to focus on in our research include home services, community services and healthcare services. Hisham et al. presented challenges in the area of knowledge for the pandemic. They concluded with re-entering the norms and standards of social distancing. Moreover, they endorsed the continuation of research after the pandemic recedes, undertaking multidisciplinary methods between fields of knowledge [18]. Chaudhury et al. explored the difference among multigenerational neighbourhoods in the metropolitan regions, concluding that older people in the higher density neighbourhoods are exposed to more traffic hazards than in the lower density neighbourhoods [19]. Azzam and Ibrahim et al. investigated the impact of COVID-19 and the global pandemic on energy sector dynamics [20]. Shiau et al. surveyed older adults to identify KPIs' importance and their degree of satisfaction relating to age-friendly transportation [21]. Metz et al. stated that it would be attractive to develop a helpful mobility framework to measure a group of benefits associated with older adults' travel and transport [22]. The age-friendly metaphor is multifactorial. For instance, Broome et al. identified that public transport can limit older people's participation in society. Therefore, focusing on public buses explained the link between bus usability and older people's health and frames existing evidence on bus usability issues [23]. COVID-19 research studies are relatively new for apparent reasons. Thus, we also reviewed literature from previous epidemic periods. Liang Fang and Zhi Dong Cao et al. [24] have constructed a real-time web system by using ArcGIS and Mashup's technology to collect and display new hotspots according to geographical location. Various models for predicting stability and MERS-CoV infection recovery have been developed by Isra Al-Turaiki et al. [25] based on Naive Bayes and J48 decision tree classification algorithms. Zhaoyang Zhang et al. [26] elaborated the epidemiological clusters by using social networking sites, collecting vital signs and social interactions. Based on this approach, we can identify and isolate the optimal bound cluster to reduce dispersion. To forecast and stem Ebola virus disease, Sareen S. Sood, S.K.; and Gupta, S.K. et al. [27] developed a cloud-based system by deploying Temporal Network Analysis (TNA) and wearable body sensor technology. Sanjay Sareen et al. [28] developed a cloud-based system for detecting and monitoring Zika virus through IoT technology's deployment.

A review of the current technological solutions state shows various digital tools to remediate the COVID-19 outbreak. Reshaped implementations of machine learning and cloud computing can contribute to the fight against the virus. Alibaba Cloud 2020 deployed machine learning and deep learning establish a modified SEIR model to predict the spreading trend of COVID-19 and evaluate the risk of infection increases in a specific region [29]. This innovative solution can provide a COVID-19 pandemic prediction report with 98% accuracy by submitting primary data such as flight information, number of new cases, number of confirmed cases, number of close contacts, contact date and number of people under quarantine. An innovative biomedical tool that could also contribute to the battle with the invisible enemy is the genomic sequence of machine modelling to forecast possible infectious reactions to various drugs or to control the spread of COVID-19 [30]. An artificial intelligence frame was also developed to assess computed tomography images identifying COVID-19 pneumonia features relative to screen infected patients [31].

## 3. Regression Models and Performance Comparison

By implementing regression analysis, which is explained in the chapter [32] and further analyzed in the article [33], we construct linear, exponential and polynomial interpretive models, as shown in Figure 1, for the cumulative infectious cases of daily reports since

31 January 2020. In addition, the least-squares method enables the calculation of the coefficients for Equations (1)–(3):

$$y = 52,205x - 2 \times 10^6 \tag{1}$$

$$y = 9784.261e^{0.055x} \tag{2}$$

$$y = 0.00003x^6 - 0.011161x^5 + 1.478908x^4 - 78.482575x^3 + 1720.083148x^2 \\ -12,628.242181x + 51,000 \tag{3}$$

where x denotes the days from the initiation of the pandemic, and y denotes the cumulative infectious cases.

Italy's cumulative infectious cases appear on the vertical axis according to the daily report (x-axis) from 31 January. Linear (1), exponential (2) and sixth-degree polynomial (3) regression modes in Figure 1, respectively, are presented by the dashed line. All models interpret 353 days of Italy's data since 31 January 2020 and are prospectively forecasting 30 days, as we present through the extension of the dashed lines.

The R-square or coefficient of determination is a statistical measure that shows the dependent variable's variance concerning the independent variable. $R^2 \in [0, 1]$ meaning that the horizontal axis values can explain 0% to 100% of the vertical axis variation.

$$R^2 = 1 - \left[ \frac{\sum_{i=1}^{n}(y_i - \hat{y}_i)^2}{(y_i - \overline{y})^2} \right] \tag{4}$$

While the numerator of (3) represents the unexplained variation, the denominator represents the total variation. Table 1 present the values of the R-squared evaluation of the three regression models, respectively.

**Table 1.** R-squared values.

| Regression Models | R-Square |
|:---:|:---:|
| Linear | $R^2 = 0.8609$ |
| Exponential | $R^2 = 0.922$ |
| 6th-degree Polynomial | $R^2 = 0.999336$ |

Regression model comparisons by the coefficient of determination $\left( R^2 \right)$ drew the conclusion that sixth-degree polynomial regression best fits the data because the value is closer to one, as shown in Table 1.

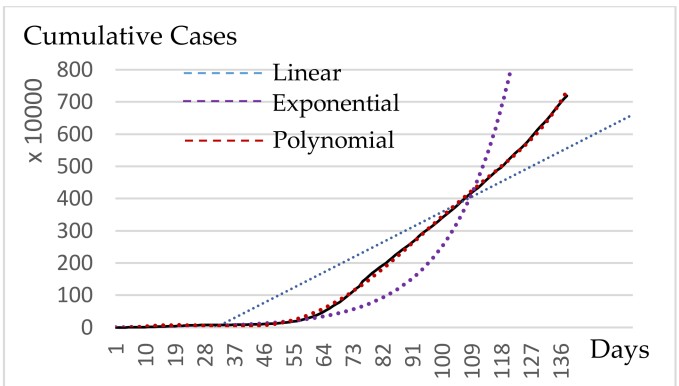

**Figure 1.** Linear, Exponential and Polynomial regression model.

## 4. Concavity and Points of Inflection

Inflexion points are defined as the change of concavity on the curve's trajectory. We refer to the intervals where the curve concaves upwards or downwards based on the second derivative sign in terms of concavity. The curve's symmetrical course around

these points motivated us to study Italy's sigmoid function from cumulative infectious cases. Before the inflexion point curve's, the orbit is concaved upwards as $F''(x) > 0$ and concaved downwards as $F''(x) < 0$. Therefore, initially, we interpret the data points by the sixth-grade polynomial function (4) that presents the best fitting, and we can calculate the second derivative. Then, the inflexion point coordinates are determined by evaluating the curve's coordinates, where the second derivative presents its root. That root will be the x-coordinate, and the value that derives from the substitution of the root on the original function will be the y-coordinate of the inflexion point.

As presented in Figure 2 we calculated the trendline with the coefficient of determination $R^2 = 0.9929$, which means a high accuracy of interpretation. After that, we determine the function of the trendline and implement the second derivative, as shown below Equations (5) and (6).

$$F(x) = -7 \times 10^{-8}x^6 + 6 \times 10^{-5}x^5 - 0.0191x^4 + 2.4617x^3 - 125.43x^2 \tag{5}$$
$$+4001.3x - 35,696$$

$$F''(x) = -2.1 \times 10^{-6}x^4 + 1.2 \times 10^{-3}x^3 - 0.2292x^2 + 14.7702x - 250.86 \tag{6}$$

$$F''(x) = 0 \Longleftrightarrow x_1 \approx 26.3226, \ x_2 \approx 77.5071,$$

$$F(x_1) = 19,183.5771, \ F(x_2) = 130,493.3966$$

The turning points estimate counts as an additional result that acts as a trademark in the pandemic. Concavity changes which are determined in Table 2 is essential because it presents the curve's furtherance until the next turning point and how the infectious cases will progress.

**Table 2.** Concavity.

| x | | $x_1$ | | $x_2$ |
|---|---|---|---|---|
| $F''(x)$ | + | | - | + |
| $F(x)$ | ⌣ | | ⌢ | ⌣ |

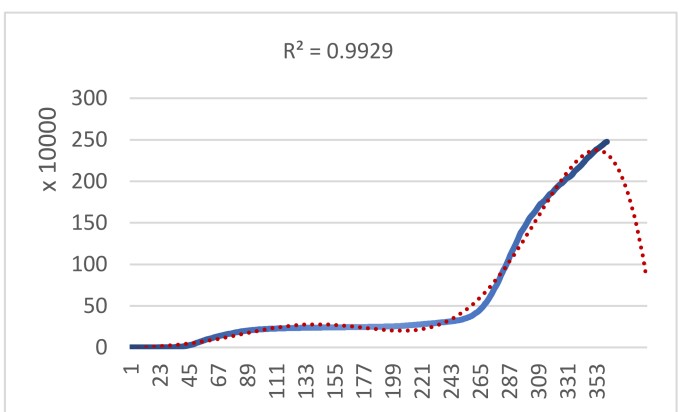

**Figure 2.** Italy's cumulative infectious cases.

## 5. Proposed Cloud Framework

In order to forecast the evolution of the COVID-19 pandemic, we propose modifying a machine learning technique deployed in a cloud-based framework. The regression models we analyzed in Section III can predict the increase and decrease in cumulative infectious cases and any other parameter that could help governments align their policy strategies against the invisible enemy. Furthermore, since late Januarys, several online open data repositories have collected various data regarding the pandemic driven from laboratories worldwide. Therefore, to forecast the progress of the unprecedented situation

with accuracy, we propose integrating repositories by using cloud computing to interpret data trends using machine learning methods.

Figure 3 illustrates the proposed model pattern, where raw data from laboratories and hospitals worldwide are routed through the network to data repositories. By modifying the Levenberg–Marquardt algorithm, a machine learning technique, we optimize the accuracy in the trend line that interprets these data sets [34]. The Levenberg [1944] and Marquardt [1963] algorithm modified the Gauss–Newton method, which provided the solution to the least square determination for nonlinear determination equation coefficients. According to section III, the polynomial that best fits the dataset has coefficients determined by the equation's minimum values (7), where x denotes the days, m denotes the amount of days from the initial outbreak and $F(x)$ is the aforementioned function.

$$F(x) = \frac{1}{2} \sum_{i=1}^{m} [f_i(x)]^2 = \frac{1}{2} \|f(x)\| \tag{7}$$

Extreme values and data noise prompted us to develop an iterative weighting strategy in order to flatten the graph of the curve and to reduce the error for greater accuracy [35]. In addition, we were reconstructing the regression model to achieve better curve adjustment and reduce outlier data distances. As shown in Equations (8)–(10), we developed three weights by composing the SoftMax function [23] and the function, which is the difference between the lengths of all the values along the y-axis from the curves of the Sigmoid function, Arc-tangent $(\tan h^{-1})$ and Hyperbolic-tangent $(\tan h)$ function in order to optimize the curve fitting process.

$$\dot{w}_i^{n+1} = \frac{e^{\left[1 - \frac{d_i^n - (1 + e^{-d_i^n})^{-1}}{\max_i d_i^n - (1 + e^{-d_i^n})^{-1}}\right]}}{\sum_i e^{\left[1 - \frac{d_i^n - (1 + e^{-d_i^n})^{-1}}{\max_i d_i^n - (1 + e^{-d_i^n})^{-1}}\right]}} \tag{8}$$

$$\ddot{w}_i^{n+1} = \frac{e^{\left[1 - \frac{d_i^n - (\frac{2}{\pi})\tanh^{-1} d_i^n}{\max_i d_i^n - (\frac{2}{\pi})\tanh^{-1} d_i^n}\right]}}{\sum_i e^{\left[1 - \frac{d_i^n - (\frac{2}{\pi})\tanh^{-1} d_i^n}{\max_i d_i^n - (\frac{2}{\pi})\tanh^{-1} d_i^n}\right]}} \tag{9}$$

$$\dddot{w}_i^{n+1} = \frac{e^{\left[1 - \frac{d_i^n - \tanh d_i^n}{\max_i d_i^n - \tanh d_i^n}\right]}}{\sum_i e^{\left[1 - \frac{d_i^n - \tanh d_i^n}{\max_i d_i^n - \tanh d_i^n}\right]}} \tag{10}$$

We divide the distances denoted by $d$ between the coordinates and the trendline by the maximum value and subtract from one. The SoftMax function standardized the results corresponding to each point. We initially provided three weights denoted by $w$ equal to one for all data points to fit the Levenberg–Marquardt Algorithm 1 curve. Then, we substitute the value calculated from (8)–(10) corresponding to every point for the next iteration $i \in \{1, 2, \ldots, n\}$. Finally, we implemented the Levenberg–Marquardt algorithm with the new weights and evaluated the curve fitting of the three methods. The sum of all weights' deviation should be lower than a threshold value to converge the algorithm.

At MATLAB's Curve Fitting Tool, we implemented a ninth-degree polynomial for curve fitting corresponding to Italy's cumulative infectious cases. As a result, we generated the curve shown in Figure 4 interpreted by (11) where x is normalized by mean 181 and std 104.4, and the coefficients' (with 95% confidence bounds) goodness of fit results are presented in Table 3.

---

**Algorithm 1. Modified Levenberg–Marquardt Algorithm**

---

**Requirements:**
  x: Input sequence of days from first reported case
  y: Input number of cases corresponding to each day in x
  t: Threshold parameter (the earliest time a failure may occur)
**Process:**
  $w_0 \leftarrow 1 * x$
  **for** iteration n from 0, step 1 **do**
    $f \leftarrow$ Levenberg Marquardt (input : $x, y, w^n$)
$$d_i \leftarrow |f(x_i) - y_i|, \ \forall i \in \mathbb{N}$$
**Apply one of the following:**
  $w_i^{n+1} \leftarrow (7)$
  $w_i^{n+1} \leftarrow (8)$
  $w_i^{n+1} \leftarrow (9)$
  **if** $\sum_i \left| w_i^n - w_i^{n+1} \right| <$ t **then**
    **break**
  **end for**
**end procedure**

---

$$F(x) = 1.088 * 10^5 x^9 + 8.952 * 10^4 x^8 - 6.961 * 10^5 x^7 - 5.891 * 10^5 x^6$$
$$+1.312 * 10^6 x^5 + 1.207 * 10^6 x^4 - 3.719 * 10^6 x^3 - 3.95 * 10^5 x^2 \qquad (11)$$
$$+5.243 * 10^4 x + 2.691 * 10^5$$

**Table 3.** Goodness of fit.

| SSE | R-Square | Adjusted R-Square | RMSE |
|---|---|---|---|
| $3.836 \times 10^{11}$ | 0.9979 | 0.9979 | $3.306 \times 10^4$ |

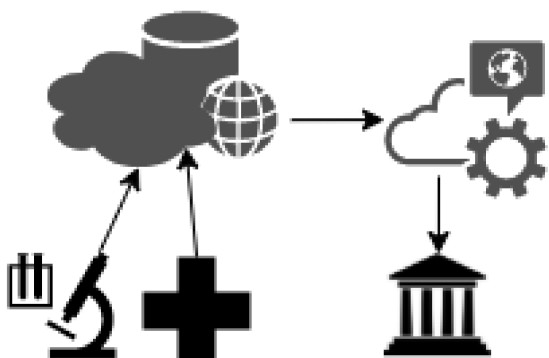

**Figure 3.** Proposed Model.

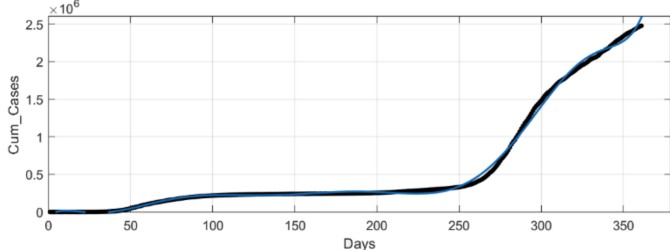

**Figure 4.** Italy's curve fitting output from MATLAB curve fitting tool.

## 6. Distribution Fitting

We present significant distribution models regarding daily new COVID-19 cases from datasets of countries that show a decline in their curve's furtherance, which was preferred. Then, we identified the best performing distributions for each patient using EasyFit Standard version 5.6.

Figures 5–9 present the distributions Johnsons $S_B$, Gen. Extreme Value, Person 6 and Degum as the best performance of goodness of fit for Italy, Czech Republic, France and Denmark, respectively, and they were evaluated by Kolmogorov Smirnov and Anderson Darling tests. Johnson $S_B$ distribution shows the best fitting performance compared to the other distributions in two countries, Italy (Figure 5) and Spain (Figure 8) [36]. From the iteratively weighted approach in section zero, the distributions fit the curve better than without weight.

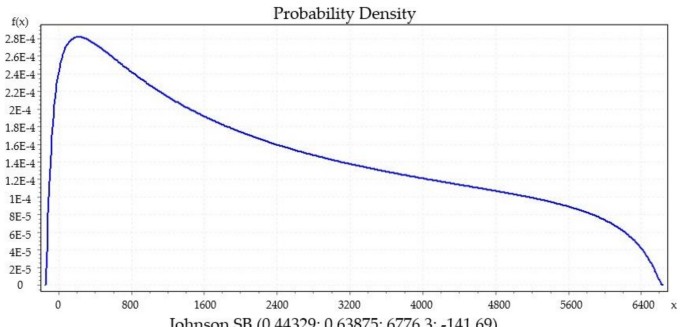

**Figure 5.** Italy.

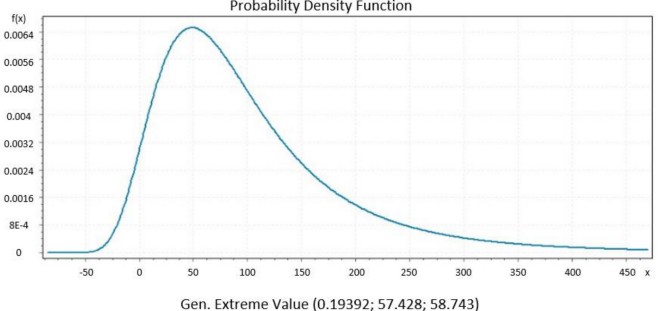

**Figure 6.** Czech Republic.

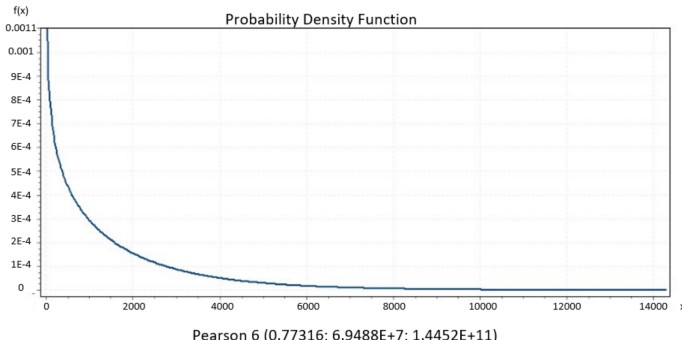

**Figure 7.** France.

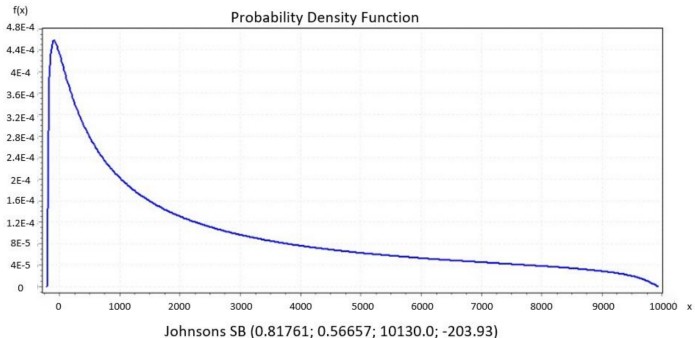

**Figure 8.** Spain.

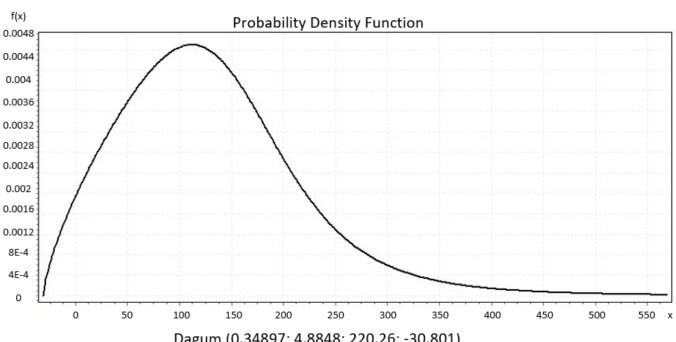

**Figure 9.** Denmark.

## 7. Personal and Health Information Protection

In order to secure the confidentiality of patients, we propose the implementation of an additional encryption process [37]. We developed a mathematical model based on data fragmentation to disintegrate the information into several cloud repositories [38]: fragmentation in terms of features distribution. Thus, their values could only be visible only by a critical enabler. We propose a necessary arithmetical analysis procedure for the decryption process: Newtons' divided difference interpolation for reconstructing the datasets. The constraints are defined as the rule for distribution among attributes. Consider A as a set of users' features and c as a set of confidentiality constraints [39]. Hence, c will be a subset of A, $c \subseteq A$, and each constraint cannot be a subset of another constraint [40]. We propose distributing and storing the subsets to database service providers (DSP) with repositories by developing polynomials. The distribution of datasets to k fragments is enable through $(k-1)$ coefficients. We denote the constant value as $a_0$, which constitutes the sensitive value of the National Identity Number, NIDN. Hence, we construct a $(k-1)$ degree polynomial as shown in the Equation (12).

$$P(x) = a_0 + a_1 x + a_2 x^2 + \cdots + a_{k-1} x^{k-1} \tag{12}$$

The database management system (DBMS) stores the secret information $x = (x_1, x_2, \ldots, x_n)$, where $n =$ fragments, and can computes the values of $P(x_i) =$ fake values of NIDN, $i = 1, 2, \ldots, k$ by substituting k of the n values from the vector x. By using Newton–Gregory's divided difference interpolation and by the knowledge of k order pairs $(x_i, P(x_i))$, $i = 1, 2, \ldots, k$, we can determine $(k-1)$ coefficients of the polynomial as well as the original value of National Identity Number NIDN corresponding to the constant $a_0$, as shown in Equation (13):

$$
\begin{aligned}
P(x) = {} & P(x_{k-1}) + \Delta_{P(x_{k-1})}(x - x_{k-1}) + \Delta^2_{\Delta P(x_{k-1})}(x - x_{k-1})(x - x_k) + \\
& \cdots + \Delta^{n-k-1}_{\Delta^{n-k-2}P(x_{k-1})} \prod_{i=k-1}^{n}(x - x_i)
\end{aligned}
\tag{13}
$$

where $\Delta_{P(x_{k-1})}$, $\Delta^2_{\Delta P(x_{k-1})}, \ldots, \Delta^{n-k-1}_{\Delta^{n-k-2} P(x_{k-1})}$ will be the first, second and $(n-k-1)^{\text{th}}$ divided differences, respectively.

*Example:*

A = {National Identity Number (NIDN), Name, Date of Birth (DoB), Mobile Number (MN), Postal Code (PC), Probability of Infection (PoI)}

$c_0$ = {NIDN} (sensitive information)

$c_1$ = {Name, DoB}

$c_2$ = {Name, MN}

$c_3$ = {Name, PC}     The names in conjunction with any other attribute are considered sensitive information

$c_4$ = {Name, PoI}

$c_5$ = {DoB, MN, PC}     The date of birth with the mobile number can infer the

$c_6$ = {DoB, MN, PoI}     identity of the users and in combination with other features is considered sensitive

An example of fragmenting the attributes which are presented in Table 4 involved in the constraints so that they are not visible together could be $f_1$ = {Name}, $f_2$ = {DoB, MN} and $f_3$ = {PC, PoI}. Fragments will be stored in three separate database service providers: database service providers 1, 2 and 3 DSP1, DSP2 and DSP3, respectively. We will develop a second-degree polynomial to share the data among database service providers DSPs, as shown in the Equation (14):

$$P(x) = a_2 x^2 + a_1 x + a_0 \tag{14}$$

where $a_0$ represents the NIDN, and the coefficients $a_1$ = (1,2,5,6,4) and $a_2$ = (7,3,2,1,9) are randomly selected. Moreover, the secret values of $x_i$, $i = 1, 2, 3$ are randomly selected and correspond to each DSP, respectively; let $x_1 = 1$, $x_2 = 2$, $x_3 = 4$. Table 5 presents the computational results of substitution to each polynomial of the coefficients and the secret values.

**Table 4.** Examples of registered data.

| NIDN | Name | DoB | MN | PC | PoI |
|---|---|---|---|---|---|
| 880,618 | Andrew | 26/03/1984 | 96,536,499 | 4529 | 80% |
| 526,548 | Nicolas | 12/05/1968 | 99,652,342 | 2324 | 95% |
| 616,636 | Jane | 13/07/1975 | 96,521,548 | 2528 | 3% |
| 844,131 | David | 25/04/1983 | 99,215,482 | 4528 | 0% |
| 321,131 | Mathew | 01/09/1950 | 99,992,272 | 5232 | 75% |

**Table 5.** Substitution results.

| NIDN | Polynomial | x = 1 | x = 2 | x = 4 |
|---|---|---|---|---|
| | P(x) | DSP$_1$ | DSP$_2$ | DSP$_3$ |
| 880,618 | $1x^2 + 7x + 880{,}618$ | 880,626 | 880,636 | 880,662 |
| 526,548 | $2x^2 + 3x + 526{,}548$ | 526,553 | 526,562 | 526,592 |
| 616,636 | $5x^2 + 2x + 616{,}636$ | 616,643 | 616,660 | 616,724 |
| 844,131 | $6x^2 + 1x + 844{,}131$ | 844,138 | 844,157 | 844,231 |
| 321,131 | $4x^2 + 9x + 321{,}131$ | 321,144 | 321,165 | 321,231 |

The fragments will be distributed as shown within the following tables (Tables 6–8) presenting an incorrect value of NIDN for each data from Table 4.

**Table 6.** Data of DSP1.

| NIDN | Name |
|---|---|
| 880,626 | Andrew |
| 526,553 | Nicolas |
| 616,643 | Jane |
| 844,138 | David |
| 321,144 | Mathew |

**Table 7.** Data of DSP2.

| NIDN | DoB | MN |
|---|---|---|
| 880,636 | 26/03/1984 | 96,536,499 |
| 526,562 | 12/05/1968 | 99,652,342 |
| 616,660 | 13/07/1975 | 96,521,548 |
| 844,157 | 25/04/1983 | 99,215,482 |
| 321,165 | 01/09/1950 | 99,992,272 |

**Table 8.** Data of DSP3.

| NIDN | PC | PoI |
|---|---|---|
| 880,662 | 4529 | 80% |
| 526,592 | 2324 | 95% |
| 616,724 | 2528 | 3% |
| 844,231 | 4528 | 0% |
| 321,231 | 5232 | 75% |

Recreating the dataset is a prerequisite for knowing the three ordered pairs $\{(x_i, P(x_i)), i = 1, 2, 3\}$, which corresponds to the three database service providers DSPs. The decryption will be achieved by employing Newton–Gregory's divided difference interpolation as shown in Table 9, from which the polynomial after reconstruction will present the original value of NIDN as the constant part of it $a_0$ [41]. Finaly, we retrieve the reconstructed Table 10.

$$P(x) = P(x_2) + \Delta_{P(x_2)}(x - x_2) + \Delta^2_{\Delta P(x_2)}(x - x_2)(x - x_3)$$

$$P(x) = 880,636 + 13(x - 2) + 1(x - 2)(x - 4)$$

$$P(x) = 880,636 + 13x - 26 + x^2 - 4x - 2x + 8$$

$$P(x) = x^2 + 7x + 880,618$$

**Table 9.** Newton–Gregory's divided difference interpolation.

| i | $x_i$ | $P(x_i)$ | $\Delta_{P(x_i)}$ | $\Delta^2_{\Delta P(x_i)}$ |
|---|---|---|---|---|
| 1 | 1 | 880,626 | | |
| | | | $\Delta_{P(x_1)} = \frac{P(x_2)-P(x_1)}{x_2-x_1} = 10$ | |
| 2 | 2 | 880,636 | | $\frac{\Delta_{P(x_2)}-\Delta_{P(x_1)}}{x_3-x_1} = 1$ |
| | | | $\Delta_{P(x_2)} = \frac{P(x_3)-P(x_2)}{x_3-x_2} = 13$ | |
| 3 | 4 | 880,662 | | |

**Table 10.** Reconstructed table.

| NIDN | Name | DoB | MN | PC | PoI |
|---|---|---|---|---|---|
| 880,618 | Andrew | 26/03/1984 | 96,536,499 | 4529 | 80% |

## 8. Mortality Rate

This section statistically analyzes the mortality rate, aiming to predict the number of deaths due to COVID-19. If we take the quantity of affirmed cases as an independent variable and the number of deaths as a dependent variable, we can find the correlation coefficient between them. We refer to implementing a strategy to evaluate a surmised linear relationship between the two continuous variables by finding the correlation. Measurement can be within the interval $[-1, 1]$ that assess an estimated direct connection between two persistent variables [42]. To this point, we utilize Student's t-test to affirm the average number of death rates with the previous quantity of cases [43]. The T-test is regularly used as a measurable strategy to examine whether the average information from an independent sample following a normal distribution is consistent or deviates from the mean estimate of a null hypothesis or whether the distinction between methods for two independent models following a normal distribution is statistically noteworthy [44].

$$r(x,y) = \frac{n\sum xy - \sum x\sum y}{\sqrt{n\sum x^2 - (\sum x)^2}\sqrt{n\sum y^2 - (\sum y)^2}} \tag{15}$$

Karl Pearson's correlation coefficient can be calculated using (15) by substituting the results derived from Table 11. We obtained the data from Italy's information as n represents the months, x represents the cumulative infectious cases and y represents the number of deaths due to COVID-19. The calculation shows that r(x,y) = 0.869185; thus, we have a high degree of positive correlation. Therefore, the mortality rate denoted by m% will increase by the increment of COVID-19 cumulative cases [45].

$$t = \frac{\overline{m} - \mu}{S/\sqrt{n}} \tag{16}$$

$$S = \sqrt{\frac{\sum(m - \overline{m})^2}{n - 1}} \tag{17}$$

In order to identify whether the data are substantial or not, we implement hypothesis testing where the null hypothesis was what Italy's average mortality rate could be according to Table 11, $\mu = 14.48\%$. From (16), (17) and Table 11, we derived $|t| = 3.45$, which shows a 0.5% level of significance and 11 degrees of freedom, confirming our null hypothesis that Italy's maximum average fatality rate could be 14.48%, where t denotes the T-test's variable, m denotes mortality, $\overline{m}$ denotes the average of the mortality, n is the total amount of the sample, S indicates the Standard deviation, and $\mu$ denotes the statistical mean.

**Table 11.** Tabulated data.

| f | x | y | m% | xy | $x^2$ | $y^2$ | $(m-\overline{m})^2$ |
|---|---|---|----|----|-------|-------|----------------------|
| 1 | 2 | 0 | 0 | 0 | 4 | 0 | 81.29 |
| 2 | 1128 | 29 | 2.57 | 32712 | 1,272,384 | 841 | 41.54 |
| 3 | 105,792 | 12,428 | 11.75 | 1,314,782,976 | $1.12 \times 10^{10}$ | $1.54 \times 10^8$ | 7.46 |
| 4 | 205,463 | 27,967 | 13.61 | 5,746,183,721 | $4.22 \times 10^{10}$ | $7.28 \times 10^8$ | 21.12 |
| 5 | 232,997 | 33,415 | 14.34 | 7,785,594,755 | $5.43 \times 10^{10}$ | $1.12 \times 10^9$ | 28.36 |
| 6 | 240,136 | 34,767 | 14.48 | 8,348,808,312 | $5.77 \times 10^{10}$ | $1.21 \times 10^9$ | 29.83 |
| 7 | 247,537 | 35,141 | 14.20 | 8,698,697,717 | $6.13 \times 10^{10}$ | $1.23 \times 10^9$ | 26.83 |
| 8 | 269,214 | 35,483 | 13.18 | 9,552,520,362 | $7.25 \times 10^{10}$ | $1.26 \times 10^9$ | 17.34 |
| 9 | 314,861 | 35,894 | 11.40 | $1.1302 \times 10^{10}$ | $9.91 \times 10^{10}$ | $1.29 \times 10^9$ | 5.68 |
| 10 | 679,430 | 38,618 | 5.68 | $2.6238 \times 10^{10}$ | $4.62 \times 10^{11}$ | $1.49 \times 10^9$ | 11.10 |
| 11 | 1,601,554 | 55,576 | 3.47 | $8.9008 \times 10^{10}$ | $2.56 \times 10^{12}$ | $3.09 \times 10^9$ | 30.76 |
| 12 | 2,047,696 | 71,925 | 3.51 | $1.4728 \times 10^{11}$ | $4.19 \times 10^{12}$ | $5.17 \times 10^9$ | 30.29 |
| n | $\sum x$ | $\sum y$ | $\sum m$ | $\sum xy$ | $\sum x^2$ | $\sum y^2$ | $\sum(m - \overline{m})^2$ |
| 12 | 5,945,808 | 381,243 | 108.19 | $3.1527 \times 10^{11}$ | $7.62 \times 10^{12}$ | $1.68 \times 10^{10}$ | 331.6104 |

## 9. Cumulative Cases vs. Daily Case Rate

The relaxation of COVID-19 lockdown measures triggered the second wave of the pandemic. As a result, most countries have reassessed, redefined, and revived COVID-19 response activities in readiness to deal with the second and potentially third wave of the outbreak. Unfortunately, in the second wave, an overblown panic occurred as cumulative cases present a contradiction concerning new cases' positivity per number of tests conducted daily. Based on that, we studied Italy's number of cumulative cases and the percentage of the PCR testing rate daily since 25 February.

As shown in Figure 10, since 16 October (day 235), we observed considerable acceleration as the curve displays rapid increasing progression. As observed, the red dashed curve represents (18), which is the trendline that is derived from the implementation of the curve adjustment to predict the extension for 20 days. Evaluation through the coefficient of determination presents that the regression predictions approximate the real data points with a fitting accuracy of $R^2 = 0.9885$.

$$y = -8 \times 10^{-8}x^6 + 8 \times 10^{-5}x^5 - 0.0249x^4 + 3.6742x^3 - 269.11x^2 + 11,479x - 82,549 \quad (18)$$

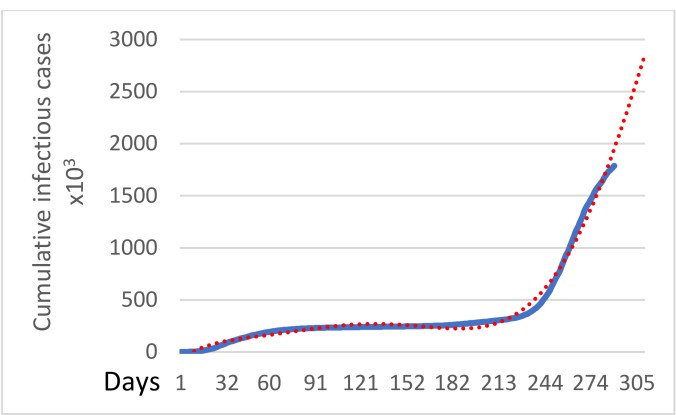

**Figure 10.** Number of cumulative cases.

We evaluated the percentage of infected cases over the number of daily conducted tests during the same period and presented the generated results graphically in Figure 11. As shown, the two schemes conclude to dissimilarities due to the curve's progress and the inferences. The curve in Figure 10 can be observed, and it changes the concavity from upwards to downwards and presents increasing progress since 06 October, day 225. More specifically, the contradiction occurs due to the rapidly growing growth of the curve in Figure 11 and the slow acceleration of growth in Figure 10. The second wave's maximum point appears to be lower than in the first wave of the pandemic. The forecast for 20 days shows that the trendline's function (18) with $R^2 = 0.866$ is decreasing within the interval [273,289] in contrast to the previous Figure 10 where the rapid increment in the interval [235, ∞) is presented.

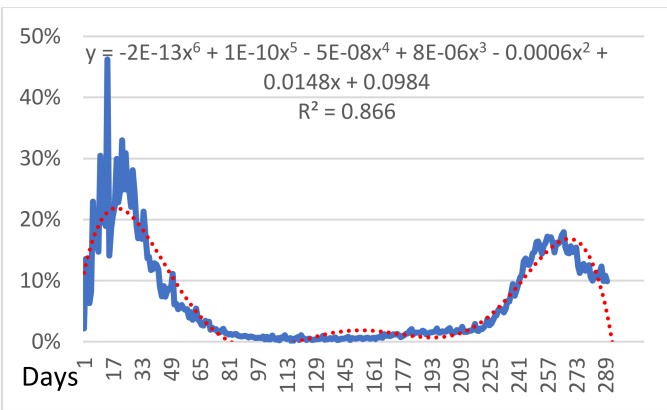

**Figure 11.** Daily infection rate.

## 10. Discussion and Conclusions

This paper proposes a modified machine learning technique to accurately interpret the data provided by fragmentation techniques by countries worldwide regarding COVID-19. The objective is to inform governments in an early stage of a pandemic situation to regulate their policies with a better strategy based on an accurate forecast model. The corresponding groups among countries utilize models to provide predictions regarding the virus's progress that could result in a contradiction. Thus, our proposed model's goal was to implement a technique that fits the regression models as best as possible to the curve. Additionally, the updated data will be driven daily to develop a new polynomial function. The goals are to estimate the upcoming turning points daily. We presented the pandemic progress hallmarks in Section 4 and forecasted the new curve's furtherance with the new polynomial. By comparing cumulative infectious cases with the daily rate of PCR testing, we concluded that it is efficient to predict the progression of a pandemic's second wave by using the daily testing rate as it effectively interprets the outbreak. According to Section 8, the mortality rate in Italy is 14.48%, as evaluated by Student's *t*-test.

As presented in Figure 12, the initial stay-at-home orders (enforced and referred to as 'lockdown') for Italy's general population began on 10 March and finished on 4 May. The second lockdown, largo, started on 26 October and was partially completed, as most restrictions were still being implemented on 4 November. Therefore, as the second pandemic wave occurred, there was an increasing trend in new cases. For this reason, if we had early lockdown strategy and if the quarantine measures had been adopted earlier, we would have avoided the unpleasant increment situation of the second wave.

The limitations from our research work include the noise that occurs in datasets due to weekends and public holidays. Due to these occurrences, the real-time data present outliers, resulting in unnecessary concavity points relative to the curves and incorrect gradients for the linear case. Henceforth, we will focus our future orientation to modify the system to exclude the corresponding data that generates this outlier by understanding each country's weekends and public holidays.

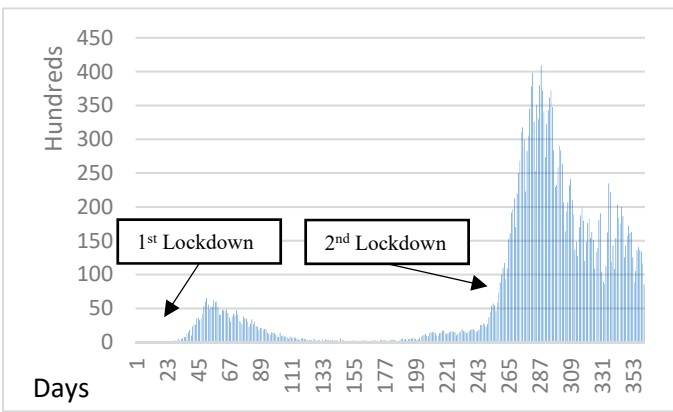

**Figure 12.** Confirmed cases.

**Author Contributions:** Writing—original draft preparation, A.A. and C.X.M.; writing—review and editing G.M., J.M.B. and E.P. All authors have read and agreed to the published version of the manuscript.

**Funding:** This research work was funded by the Smart and Health Ageing through People Engaging in supporting Systems SHAPES project, which has received funding from the European Union's Horizon 2020 research and innovation programme under grant agreement No 857159. Parts of this work were supported by the Ambient Assisted Living (AAL) project vINCI: "Clinically-validated INtegrated Support for Assistive Care and Lifestyle Improvement: The Human Link", funded by Cyprus Research and Innovation Foundation in Cyprus under the AAL framework with Grant Nr. vINCI /P2P/AAL/0217/0016.

**Data Availability Statement:** No new data were created or analyzed in this study. Data sharing is not applicable to this article.

**Conflicts of Interest:** The authors have no conflict of interest to declare. All co-authors have seen and agree with the contents of the manuscript and there is no financial interest to report. We certify that the submission is original work and is not under review at any other publication.

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
