# Peer review of "Evaluation of the COVID-19 Era by Using Machine Learning and Interpretation of Confidential Dataset"

_electronics, doi:10.3390/electronics10232910_

Round 1

Reviewer 1 Report

The paper proposes a modified machine learning approach for properly interpreting COVID-19 data given by governments throughout the world using fragmentation techniques. The goal is to alert governments early in a pandemic crisis so that they may better manage their policies based on an accurate forecast model. The objective of the suggested model was to design a strategy that matched regression models to the curve as closely as feasible also using the updated to create a new polynomial function on a daily basis. The authors discussed the characteristics of pandemic development and used a new polynomial formulation to anticipate the new curve's progression. Comparing cumulative infected cases to the daily rate of PCR testing, it involves that using the daily testing rate to forecast the progression of a pandemic's second wave is useful because it effectively interprets the epidemic. The results put in evidence the nasty increase situation of the second wave if an early shutdown approach and quarantine procedures are used.

The article is well structured and written in a clear way. The reasons and objectives that the authors set themselves with their proposal are understandable and the proposal deserves to be published. To allow a better understanding of the contents I suggest to the authors to:

  • in section 3: Please recall the regression analysis, albeit briefly, by inserting some reference for the reader who would like to review the specifics of the topic
  • arrange references: they are not displayed correctly in the text
  • again in paragraph 3: I would try to replace the three graphs with a single graph with colored curves. This would make it easier to make a comparison
  • it would be useful if the authors highlight the limitations of the studies and possible future developments

Author Response

Dear Reviewer,

We would like to thank you and kindly inform you that we have followed your comments and your suggestions towards preparing the revised version of our manuscript.

Please find for your consideration the current version that satisfies the requirements. The quality of our research work has been improved, according to your valuable/constructive comments.

Responses:

  • in section 3: Please recall the regression analysis, albeit briefly, by inserting some reference for the reader who would like to review the specifics of the topic

We would like to thank you for the suggestion. We briefly added two references [28] and [29] in section 3 for the reader who would like to review the specifics of the topic.

  • arrange references: they are not displayed correctly in the text

We would like to thank you for the comment and kindly inform you that we have addressed it accordingly.

  • again in paragraph 3: I would try to replace the three graphs with a single graph with colored curves. This would make it easier to make a comparison

We would like to thank you for the recommendation and kindly inform you that we enable the figure accordingly (figure 1.)

  • it would be useful if the authors highlight the limitations of the studies and possible future developments

We would like to thank you for the suggestion and kindly inform you that we enable a paragraph in the Concussion section that highlights the limitations of our studies and our future orientation to approach the solution.

Please accept our appreciation for your editorial efforts, and do not hesitate to contact us if our side requires further action.

An acknowledgements section hosts our thanks to the reviewers for their valuable comments, which helped us significantly improve our paper’s presentation and our research work quality.

Yours sincerely,

(on behalf of all the authors of our work)

Andreas Andreou

Reviewer 2 Report

This paper reports a case study using machine learning to analyze the Covid-19 related dataset. Though the authors showed some data based analysis along with the modeling research, the novelty of this work is rather limited. None of the conclusion, content, method, results is new. This looks like a simple computational experimental report instead of a scientific paper that can make contribution to the community. Therefore, I cannot recommend that this paper is considered for publication.

Author Response

Dear Reviewer,

We would like thank you and kindly inform you that we have followed the reviewers’ comments and their suggestions towards preparing the revised version of our manuscript.

Please find for your consideration the current version that satisfies the reviewers’ requirements. The quality of our research work has been improved, according to their valuable/constructive comments.

An acknowledgements section hosts our thanks to the reviewers for their valuable comments, which helped us significantly improve our paper’s presentation and our research work quality.

Please accept our appreciation for your editorial efforts, and do not hesitate to contact us if our side requires further action.

Yours sincerely,

(on behalf of all the authors of our work)

Andreas Andreou

Reviewer 3 Report

Authors introduced an evaluation of the COVID-19 Era using Machine Learning. The idea and flow of the paper are good. However, I will recommend  following points to be added before publication.

  • First paragraph of the related work to be changed.
  •  Contributions to be added 
  •  More results to be added to validate the idea.
  •  I will recommend to add following work either in related work or future work  Razaque, Abdul, Bakhytzhan Valiyev, Bandar Alotaibi, Munif Alotaibi, Saule Amanzholova, and Aziz Alotaibi. "Influence of COVID-19 Epidemic on Dark Web Contents." (2021).,,,,,
  • Variables for all of the equations should be declared otherwise it will be harder for reader to understand.
  • Units are missing in the most of graphs on the X-axis and Y-axis
  • Quality of the figures should be improved.
  • More references should be added in the introduction part because only 2 references in the introduction part is not enough.
  • Research question should be raised in the introduction part.
  • Conclusion part is weak that should specify what has been achieved.

Author Response

Dear Reviewer,

We would like to thank you and kindly inform you that we have followed the reviewers’ comments and their suggestions towards preparing the revised version of our manuscript.

Please find for your consideration the current version that satisfies the reviewers’ requirements. The quality of our research work has been improved, according to their valuable/constructive comments.

Response:

  • First paragraph of the related work to be changed.

We would like to thank you for the recommendation and kindly inform you that the first paragraph has been changed accordingly

  •  I will recommend to add following work either in related work or future work  Razaque, Abdul, Bakhytzhan Valiyev, Bandar Alotaibi, Munif Alotaibi, Saule Amanzholova, and Aziz Alotaibi. "Influence of COVID-19 Epidemic on Dark Web Contents." (2021).,,,,,

We would like to thank you for the recommendation and kindly inform you that we insert the corresponding research work.

  • Units are missing in the most of graphs on the X-axis and Y-axis

We would like to thank you for the suggestion and kindly inform you that we enable the corresponding units as required

  • Quality of the figures should be improved.

We would like to thank you for the suggestion and kindly inform you that we elaborate the figures accordingly. In the case of graphical figures, they derived from Microsoft Excel and cannot be updated further.

  • More references should be added in the introduction part because only 2 references in the introduction part is not enough.

We would like to thank you for the suggestion and kindly inform you that we thoroughly elaborate the introductory section by including more references according to your valuable recommendation.

  • Conclusion part is weak that should specify what has been achieved.

We include an additional paragraph to the conclusion section that  highlights the limitations of the research and how will be addressed in the future.

An acknowledgements section hosts our thanks to the reviewers for their valuable comments, which helped us significantly improve our paper’s presentation and our research work quality.

Please accept our appreciation for your editorial efforts, and do not hesitate to contact us if our side requires further action.

Yours sincerely,

(on behalf of all the authors of our work)

Andreas Andreou

Round 2

Reviewer 2 Report

It can be accepted now.

Author Response

We appreciate the reviewer’s comment, who expresses satisfaction with the current version of our paper.

Reviewer 3 Report

The authors tried their best to respond my comments, but they partially responded. However, there are still  having many deficiencies in the article but want them at least to fix following  comments:

  • Razaque, Valiyev, Alotaibi, Amanzholova and Alotaibi et. al >>> should be cited as:     Razaque et al.
  • Quality of Figures 4-8 is low. 
  • Variables used for equations 3, 6-9 are not defined.
  • Equation-10 is missing
  • what are they going to prove in equation-11?
  • There is no numbering of equations 12-14
  • Variables are not defined for equations 15-16
  • Authors again missed to give the units for X-axis and Y-axis. Without giving unit names, it will be harder for reader to understand validity of the results.
  • Most of the acronyms are not defined and I highlight few here: DSP1, DSP2, and DSP3. Kindly fix all of the acronyms in entire article.
  • what are they going to prove in equation-17?
  • On line 153, there is no reference for Xing-Yi et al
  • There are a lot of typos and grammatical mistakes that must be fixed

For example   on line 443 Concussion>>> Conclusion: { Concussion is a traumatic brain injury that affects your brain function].

Figure. 4, Figure. 5, Figure. 6, Figure. 7, and Figure. 8 presents>>>>Figure. 4, Figure. 5, Figure. 6, Figure. 7, and Figure. 8 present

Additional comments: There is need to improve the flow of writing because most of the paragraphs are incomplete in the article and I highly recommend re-writing the the article and it requires extensive English editing before submitting to Journal. They can send the article to www.aje.com for English Editing.

Author Response

November 18th 2021

Dear Reviewer,

We would like to thank you and kindly inform you that we have followed your comments and suggestions towards preparing the revised version of our manuscript.

Please find for your consideration the current version that satisfies the requirements. The quality of our research work has been improved, according to your valuable/constructive comments.

Reviewers’ comments:

  1. Razaque, Valiyev, Alotaibi, Amanzholova and Alotaibi et. al >>> should be cited as: Razaque et al.

Ans.:

We would like to thank you for the comment and kindly inform you that we quote it according to your request.

  1. Quality of Figures 4-8 is low.

Ans.:

We would like to thank you for the comment and to inform you that we have upgraded the quality of the figures 4-8.

  1. Variables used for equations 3, 6-9 are not defined.

Ans.:

We would like to thank you for the comment and kindly inform you that we have define all the variables used within the aforementioned equations.

  1. Equation-10 is missing.

Ans.:

We would like to thank you for the comment and to inform you that we have corrected the missing number in the corresponding equation.

  1. what are they going to prove in equation-11?

Ans.:

Thank you for the question. We would like to inform you that equation 11 which now changed into 10 is a 9th degree trendline for the curve, generated by MATLAB’s Curve Fitting Tool.

  1. what are they going to prove in equation-11?

Ans.:

We would like to thank you for the comment and to inform you that we have corrected the missing number in the corresponding equation.

  1. There is no numbering of equations 12-14

Ans.:

We would like to thank you for the comment and to inform you that we have enabled numbering in the corresponding equation which now are 11-13.

  1. Variables are not defined for equations 15-16

Ans.:

We would like to thank you for the comment and to inform you that we define the variables of the aforementioned equations.

  1. Authors again missed to give the units for X-axis and Y-axis. Without giving unit names, it will be harder for reader to understand validity of the results.

Ans.:

We would like to thank you for the comment and to inform you that we enable the units for X-axis and Y-axis. Thank you once again it is true that it enables the readability of the graph.

  1. Most of the acronyms are not defined and I highlight few here: DSP1, DSP2, and DSP3. Kindly fix all of the acronyms in entire article.

Ans.:

We would like to thank you for the comment and to inform you that we define the notations. Please kindly see at the line 362 where most of them are defined.

  1. what are they going to prove in equation-17?

Ans.:

We would like to thank you for the question and respond that the forecast for 20 days shows that the trendline’s function (17) with R2=0.866 is decreasing within the interval [273,289] in contrast to the previous Figure 12 where presented rapid increment in the interval [235, ∞).

  1. On line 153, there is no reference for Xing-Yi et al

Ans.:

We would like to thank you for the comment and to inform you that we have corrected ref. 20.

  1. There are a lot of typos and grammatical mistakes that must be fixed. For example, on line 443 Concussion>>> Conclusion: {Concussion is a traumatic brain injury that affects your brain function]. Figure. 4, Figure. 5, Figure. 6, Figure. 7, and Figure. 8 presents>>>>Figure. 4, Figure. 5, Figure. 6, Figure. 7, and Figure. 8 present

Ans.:

We would like to thank you for the comment and to inform you that we have thoroughly elaborate the text towards correcting the majority of the typographic mistakes. Also, we would like to thank you for indicating this important typo in line 443 as well as in line 327.

Please accept our appreciation for your editorial efforts, and do not hesitate to contact us if our side requires further action.

Yours sincerely,

(on behalf of all the authors of our work)

Andreas Andreou

Round 3

Reviewer 3 Report

The authors addressed my all concerns